# Overexpression of *pPLAIIIγ* in Arabidopsis Reduced Xylem Lignification of Stem by Regulating Peroxidases

**DOI:** 10.3390/plants11020200

**Published:** 2022-01-13

**Authors:** Jin Hoon Jang, Hae Seong Seo, Ok Ran Lee

**Affiliations:** 1Department of Applied Plant Science, College of Agriculture and Life Science, Chonnam National University, Gwangju 61186, Korea; Jinhun92@naver.com (J.H.J.); ss4540@naver.com (H.S.S.); 2AgriBio Institute of Climate Change Management, Chonnam National University, Gwangju 61186, Korea; 3Interdisciplinary Program in IT-Bio Convergence System, Chonnam National University, Gwangju 61186, Korea

**Keywords:** Arabidopsis, phospholipase, lignin, peroxidase, xylem, *pPLAIIIγ*

## Abstract

Patatin-related phospholipases A (*pPLAs*) are a group of plant-specific acyl lipid hydrolases that share less homology with phospholipases than that observed in other organisms. Out of the three known subfamilies (*pPLAI*, *pPLAII*, and *pPLAIII*), the *pPLAIII* member of genes is particularly known for modifying the cell wall structure, resulting in less lignin content. Overexpression of *pPLAIII**α* and ginseng-derived *Pg**pPLAIIIβ* in Arabidopsis and hybrid poplar was reported to reduce the lignin content. Lignin is a complex racemic phenolic heteropolymer that forms the key structural material supporting most of the tissues in plants and plays an important role in the adaptive strategies of vascular plants. However, lignin exerts a negative impact on the utilization of plant biomass in the paper and pulp industry, forage digestibility, textile industry, and production of biofuel. Therefore, the overexpression of *p**PLAIII**γ* in Arabidopsis was analyzed in this study. This overexpression led to the formation of dwarf plants with altered anisotropic growth and reduced lignification of the stem. Transcript levels of lignin biosynthesis-related genes, as well as lignin-specific transcription factors, decreased. Peroxidase-mediated oxidation of monolignols occurs in the final stage of lignin polymerization. Two secondary cell wall-specific peroxidases were downregulated following lowered H_2_O_2_ levels, which suggests a functional role of peroxidase in the reduction of lignification by *p**PLAIII**γ* when overexpressed in Arabidopsis.

## 1. Introduction

The global demand for plant resources for food and fuel consumption has been estimated to increase up to 50% [1]. Thus, engineering the raw plant materials for valuable biomass is necessary to meet the worldwide energy demand. The cell wall of plant biomass is primarily composed of three biopolymers including cellulose, hemicellulose, and lignin. It also includes other minor components such as organic acids, tannins, proteins, and secondary metabolites. The composition of lignocellulosic material varies between different plant species, and sometimes becomes an impediment in developing biomass engineering techniques for bioethanol production [2]. The pretreatment step to separate cellulose from lignin during bioethanol production using plant cell walls is costly. Although lignin is a class of complex aromatic biopolymers that provides fundamental structural support to plant cell walls, the presence of lignin is considered the major cause of biomass recalcitrance. Hence, the level or composition of lignin has been widely studied to improve biomass digestibility and reduce recalcitrance during bioengineering. Plant biomass with less lignin content has been engineered by downregulation or knocking out genes involved in lignin biosynthesis or transcription factors controlling lignin levels [2]. Our group has also previously reported that the genetic engineering of patatin-related phospholipase *pPLAIIIα* from Arabidopsis and *pPLAIIIβ* from ginseng by overexpression in Arabidopsis and hybrid poplar reduced the total content of lignin and xylem lignification [3,4,5]. However, the functional roles of other close homologs *pPLAIII**γ* and *pPLAIIIδ* focused on lignification were not yet characterized.

The Arabidopsis phospholipase (PLA) superfamily has been classified into four groups [6,7], including phosphatidylcholine (PC) hydrolyzing PLA_1_s, phosphatidic acid (PA)-preferring PLA_1_s, secretory low molecular weight PLA_2_s (sPLA_2_s), and patatin-related phospholipases (*pPLAs*). Phospholipases are complex and important enzymes that are involved in many physiological processes in plants, such as lipid biosynthesis and metabolism, stress responses, cellular signal transduction, cell growth regulation, and membrane homeostasis [6,7]. PLAs act on a phospho- or a galacto-glycerolipid to release free fatty acids and lysophospholipids [8,9]. Overexpression of *pPLAIIIβ* (*pPLAIIIβOE*) from native Arabidopsis plant resulted in an increase in all phospholipids and galactolipids, and a decrease in cellulose content without altering lignin content [10]. One of its homologs, *p**PLAIIIα**,* also displayed similar stunted phenotypes when overexpressed [11]. However, *p**PLAIIIαOE* reduced the cellular lipid species to a greater extent than that of the wild-type in Arabidopsis and rice [11,12]. Contrary to the action of *pPLAIIIβOE* that reduced the cellulose content, *pPLAIIIαOE* only reduced lignin content without altering cellulose content, with a reduced level of H_2_O_2_ compared to the control [11]. These data suggest that distinct and redundant functional roles exist among four isoforms of *pPLAIII* genes.

Oxidation of monolignols is required in the final stage of lignin polymerization, and peroxidases (PRXs) are involved in the random cross-linking to facilitate lignin polymerization [13]. Two PRXs, PRX64 and PRX72, function in the control of the spatiotemporal deposition of lignin during different developmental stages by using differentially distributed oxidative substrates, such as H_2_O_2_ [13]. Thus, it is important to understand the functional roles of *pPLAIII* isoforms in the regulation of lignification of the stem. Therefore, the regulatory role of *pPLAIII**γ* was analyzed by overexpression in native Arabidopsis to understand the contribution of each of its isoforms to lignin biosynthesis. Spatial and temporal *pPLAIII**γ* expression, phloroglucinol lignin staining, and direct lignin quantification showed that *pPLAIII**γ* is also involved in reducing lignin content. Decreased levels of H_2_O_2_ and downregulation of both transcripts of *PRX64* and *PRX72* further showed a novel pathway of *pPLAIII* genes in stem lignification.

## 2. Results

### 2.1. High Expression of PropPLAIIIγ::GUS in Xylem and Phloem Cells

Our previous study showed that overexpression of *pPLAIIIα* [5] and *PgpPLAIII**β* in the Arabidopsis system [3,5] reduced lignin content in the stem of each transformant. Analysis of spatial expression patterns of promoter::*GUS* (*β-glucuronidase*) fusion, *Pro**pPLAIII**α*::*GUS* transformants showed that there was an increased gene expression in the xylem and phloem of cross-sectioned stems [11]. In this study, independent *Pro**pPLAIII**γ*::*GUS* transformants were generated, and images were taken throughout different developmental stages to gain insights into the temporal and spatial distribution patterns of *pPLAIIIγ* transcripts in the stem (Figure 1). The GUS reporter gene was found to be expressed in most organs, including the seedling, inflorescence, flower, stem, leaf, and root (Figure 1A–H). These temporal and spatial GUS expression patterns provide additional information in addition to the quantitative real-time PCR results of the root [10]. The GUS staining became stronger in the hypocotyls of etiolated seedlings compared to those grown in light conditions (Figure 1C). *PropPLAIIIγ*::*GUS* was also highly expressed in roots and root hairs with greater restriction in the vasculature (Figure 1D,E). Intense staining was observed at the junction of the silique and stigma and throughout the vein of sepals in the floral organs (Figure 1G). Cross-sections of the apical and basal part of the stem show the expression of *PropPLAIIIγ*::*GUS* in xylem and phloem (Figure 1H), which is similar to that of our previous study on *Pro**pPLAIII**α*::*GUS* [11]. These GUS reporter gene expression patterns indicate the possible role of *p**PLAIII**γ* in stem lignification.

### 2.2. Overexpression of pPLAIIIγ Reduced Plant Height

All the overexpression lines of *pPLAIIIαOE* [5,11], *pPLAIIIβOE* [3,4,10], and *pPLAIIIδOE* [14] displayed overall stunted growth patterns with reduced plant height and radially expanded cell elongation. Reduced longitudinal growth patterns, in which the anisotropic cell expansion was disrupted, focused on floral organs and leaves and were very recently published by overexpression of *pPLAIIIγ* (*pPLAIIIγOE*) [15]. However, the phenotypic and biochemical characteristics of *pPLAIIIγOE* in stem elongation and lignification were not yet characterized. Two independent transgenic lines under the 35S promoter were chosen for further work out of several transgenic lines that followed the Mendelian law of segregation. Line numbers 5 and 14 were quantified by qPCR and were seen to have 2600-fold and 5800-fold more transcripts, respectively, compared to that of 4-week-old vegetative leaves of Col-0 wild-type (Figure 2A). Interestingly, the number of *p**PLAIII**γ* and *pPLAIIIδ* transcripts were inversely proportional to each other. However, the mRNA levels of *pPLAIIIα* and *pPLAIIIβ* were not altered (Figure 2A), indicating that the function of *p**PLAIII**γ* might be independent to that of both *pPLAIIIα* and *pPLAIIIβ*. Two selected transgenic *p**PLAIII**γOE* lines displayed intermediate and severely dwarfed plant height (Figure 2B). Isotropic growth patterns [15] were more distinctively observed in the stem (Figure 2C), which ultimately led to shorter plant height in OE lines than that in the Col-0 lines.

### 2.3. Overexpression of pPLAIIIγ Reduced Lignin Content in Arabidopsis

Our previous study reported that the overexpression of *pPLAIIIα* from Arabidopsis [5] and *PgpPLAIIIβ* from ginseng [3,4] reduced the lignification of xylem in Arabidopsis and hybrid poplars without altering cellulose content. However, overexpression of *pPLAIIIβOE* in Arabidopsis resulted in a reduction in cellulose content [10]. Therefore, tissue staining and direct measurement of lignin were performed using *pPLAIII**γOE* lines to clarify this difference in the modification of cell wall composition by these homologs. Lignin staining and quantification were performed, adding one more transgenic line to obtain more robust confirmatory results.

Treatment with phloroglucinol-HCl stain led to a pink color due to the reaction with cinnamaldehyde end-groups of lignin. Reduced lignin staining was observed in xylem cell layers of *pPLAIIIγOE* lines compared to the control (Figure 3A). The transcript levels of *pPLAIII**γ* were quantified using the stems used for lignin staining and quantification. Line #1 showed an even higher expression than line #14 (Figure 3B). This indicated that lines #1 and #14 showed an almost equally high expression of the *pPLAIII**γ* gene compared to line #5. Samples used for lignin staining and quantification and cellulose analysis are represented in Figure 3C. Consistent with the results for the levels of transcripts observed for each line (OE#1 > OE#14 >> OE#5), where OE#1 showed the highest expression, lignin staining of cross-sections was severely decreased in the OE lines, showing an inverse relationship with the degree of gene expression. Direct quantification of lignin content by acetyl bromide further showed significantly decreased lignin content in two highly expressing OE lines (Figure 3D), which further confirmed the results of phloroglucinol staining (Figure 3A). The content of cellulose, the other major cell wall component, was not altered (Figure 3E), although relevant transcripts involved in cellulose biosynthesis [16] and cellulose production [17] were slightly upregulated (Appendix A). Overall, the results strongly indicate that the overexpression of *pPLAIII**γ* in the stem ultimately reduced the lignin content.

### 2.4. Lignin-Specific Transcription Factors Are Downregulated in the Stem of pPLAIIIγOE

Two key transcription factors, *MYB58* and *MYB63*, involved in lignin biosynthesis [18] were significantly reduced in OE lines (Figure 4A). Relevant monolignol biosynthetic genes, including *PAL1* (phenylalanine ammonia-lyase 1), *4CL* (4-coumarate-CoA ligase)*, HCT* (hydroxycinnamoyl-CoA: shikimate/quinate hydroxycinnamoyl transferase)*, COMT* (caffeic acid *O*-methyltransferase), *CCR1* (cinnamoyl-CoA reductases 1)*,* and *F5H* (ferulic acid 5-hydroxylase, also named *CYP84A4*) were all significantly downregulated in the stem of all the transgenic plants overexpressing *pPLAIII**γ* (Figure 4B). This suggests that *pPLAIII**γOE* is involved in the negative regulation of lignification in Arabidopsis.

### 2.5. Transcripts Encoding Oxidative Enzyme Are Altered in pPLAIIIγOE

Monolignols, which are lignin precursors, are synthesized within the cytosol and subsequently secreted to the apoplastic cell wall where they are oxidized, which initiates random cross-linking to form lignin polymers. Secreted enzymes, namely peroxidases (PRXs), are known to facilitate lignin polymerization by oxidizing lignin monolignols using oxidative substrates such as hydrogen peroxide (H_2_O_2_). It was reported recently that overexpression of *pPLAIIIα* led to reduced levels of H_2_O_2_ in rosette leaves with delayed senescence [11]. Therefore, the levels of H_2_O_2_ were monitored in two independent *pPLAIII**γOE*. In situ detection of H_2_O_2_ was performed by staining with 3,3-diaminobenzidine (DAB) [19]. DAB was oxidized by H_2_O_2_ in the presence of heme-containing oxidoreductase to generate a dark brown precipitate. Cross-sections of each stem were visualized after vacuum infiltration to observe the precise localization of DAB staining. The levels of H_2_O_2_ decreased due to the overexpression of *pPLAIIIγ* compared to those of the Col-0 and empty vector controls (Figure 5A).

Lignification appears to be tightly regulated by localized oxidative enzymes and ROS accumulation. PRX64 is expressed in stems and are proven to be involved in lignin polymerization [20]. Additionally, PRX64, PRX71, and PRX72 were identified as potential enzymes involved in the lignification of the stem out of 73 PRXs identified in Arabidopsis [13]. PRX72 localize to the thick secondary wall of xylem vessels and fiber cells, while PRX64 localizes to the cell corners and middle lamella of fiber cells [13]. Quantitative gene expression analysis was performed to further confirm the function of *pPLAIIIγOE* in lignification through the regulation of genes encoding oxidative enzymes (Figure 5B). The two peroxidase genes, *PRX64* and *PRX72*, were significantly downregulated in two *pPLAIII**γOE* lines (Figure 5B). Altogether, the results suggest that overexpression of *pPLAIII**γ* is involved in reducing stem lignification via the downregulation of *PRX64* and *PRX72*.

## 3. Discussion

Lignin deposition is important for the modification of the major cell wall structure depending on different developmental stages and environmental stimuli. Lignin constitutes between 18% and 35% of the overall plant biomass [2]. It provides mechanical strength to the cell walls for upright growth and efficient water transport via xylem vessels. In addition to this fundamental importance of lignin, there is a biotechnological incentive to reduce lignin formation as it is a major obstacle to the utilization of cellulosic biofuels [21]. A recent report showed that the overexpression of *pPLAIIIα* resulted in less lignification of the stem [11]. Thus, further work focused on revealing whether the reduction in lignin by overexpression of *pPLAIIIs* is a general feature is demanded. Spatial gene expression of *pPLAIII**γ* in xylem vessels and interfascicular cells (Figure 1H) was the first confirmative clue to uncover its possible role in stem lignification [3,4,5,11]. Constitutive overexpression of *pPLAIII**γ* resulted in severe dwarfed plant height, and the severity of this dwarfism is positively correlated with the gene expression (Figure 2). Stunted plant growth is now proven to be a general phenotype of the overexpression of *pPLAIIIs* [3,4,5,10,11,12,14]. However, the biological mechanism of the formation of dwarf plants seems to be differentially regulated since the modified lipid species and fatty acids generated by overexpression of different isoforms of *pPLAIIIs* are different [10,11,12].

*MYB58* and *MYB63*, which are well-known lignin-specific transcription factors, are significantly downregulated by 45% and 40% on average, respectively, in *pPLAIII**γ* overexpression lines (Figure 4A). Expression levels of monolignol biosynthesis genes such as *PAL1*, C*OMT*, *4CL*, *HCT*, *CCR1*, and *F5H* were also significantly reduced from 14% to 36% on average (Figure 4B). The quantification of lignin (Figure 3D) corroborated the results of the phloroglucinol-HCl staining (Figure 3A) based on the levels of transcripts (Figure 3B). This suggested that *pPLAIII**γOE* also functions in the negative regulation of lignin biosynthesis at the transcription level in a manner similar to that of *pPLAIIIαOE* [5].

The final stage of lignin polymerization requires oxidation of monolignols by secreted oxidative enzymes, namely class-III peroxidases (PRXs) that form cross-linkages to produce the lignin polymers. Peroxidases are heme-containing oxidoreductases that require H_2_O_2_ to oxidize phenolic compounds [22]. H_2_O_2_ levels control lignin deposition [13]. The cellular level of H_2_O_2_ was reduced by the overexpression of the patatin-related phospholipase *pPLAIIIα* [11]. Similarly, overexpression of *pPLAIII**γ*, an isoform of *pPLAIIIα,* was also found to reduce the level of H_2_O_2_ by DAB staining in the present study (Figure 5A). The involvement of oxidative enzymes in lignification and the observed result of reduced H_2_O_2_ by *pPLAIII**γOE* strongly suggests that PRXs play a major role in stem lignification upon *pPLAIII**γOE.* Out of the several identified *PRX* genes involved in stem lignification [13], the transcripts of *PRX64* and *PRX72* were observed to be 16% and 21% downregulated, respectively (Figure 5B). PRX72 localize to the secondary cell wall of xylem vessels and fibers, while PRX64 localizes to the fiber cell corners [13]. The gene encoding PRX71 that localized to the same site with PRX64 did not show altered transcript levels (Appendix A). PRX42 and PRX52 were reported to be localized in nonlignified tissues [13], and the relevant gene expressions of *PRX42* and *PRX52* were not dramatically altered (Appendix A). The transcript levels of two distinct genes encoding the oxidative enzymes *PRX64* and *PRX72* were significantly decreased (Figure 5B). These results suggest that reduction of lignin by *pPLAIII**γOE* occurs due to less oxidation of monolignols that are catalyzed by PRX64 and PRX72 in the stem. Reduced transcript levels of *PRX64* and *PRX72* seem to be playing important roles in the reduction of stem lignification by *pPLAIII**γOE* due to the lower availability of H_2_O_2_ as a substrate.

## 4. Materials and Methods

### 4.1. Plant Materials and Growth Conditions

The *Arabidopsis thaliana* (L.) Heynh. ecotype Columbia (Col) was used for all wild-type and overexpression backgrounds in this study. Seeds were sown on half-strength MS medium (M0222, Duchefa Biochemie, Haarlem, The Netherlands) containing 1% sucrose and 0.8% agarose under long-day photoperiod conditions of 16 h light/8 h dark at 23 °C. Germinated seedlings were transplanted to a soil mixture containing autoclaved soil, vermiculite, and pearlite (3:2:1 *v*/*v*/*v*).

### 4.2. Transgenic Construct and in Planta Transformation

The full-length genomic DNA sequence of *pPLAIII**γ* from Arabidopsis was cloned into the modified pCAMBIA1390 vector [11] under the control of the cauliflower mosaic virus 35S promoter and yellow fluorescence protein (YFP) C-terminal tagging. The full-length genomic fragment of *pPLAIII**γ* was amplified using the following primers: 5′-TC GGT ACC GTC TAA AAG CTA ACG ATT-3′ and 5′-GG CCT AGG TCT ATC TTT AGA TAT GAG-3′. The construct was confirmed by nucleotide sequencing and transformed *in planta* into Arabidopsis using *Agrobacterium tumefaciens* C58C1 (pMP90) [23] by the floral dipping method [24]. The *PropPLAIIIγ*::*GUS* fusion construct was generated by cloning the upstream intergenic region containing the full promoter sequence of *pPLAIIIγ*. The promoter region was amplified using the following primers: 5′-TC CTG CAG ATT CAC TTT TCT TCT TGT-3′ (forward) and 5′-TC GTC GAC AAT CGT TAG CTT TTA GAC-3′ (reverse). The amplified and purified PCR product was subsequently cloned into a pCAMBIA1390 vector containing a *gusA* reporter gene. Each transformant was selected on hygromycin-containing plates (50 μg/mL). Homozygous plants following Mendelian inheritance were selected on antibiotic plates for further analyses.

### 4.3. β-Glucuronidase (GUS) Histochemical Analysis and Phloroglucinol-HCl Staining

*PropPLAIII**γ*::*GUS* expressing transgenics from the seedling to the adult stage were used to visualize GUS activity. Histochemical GUS staining was performed by incubating *Pro**pPLAIII**γ*::*GUS* transgenics in staining buffer containing 1 mM 5-bromo-4-chloro-3-indolyl-β-D-glucuronic acid cyclohexylammonium salt (X-Gluc, Duchefa Biocheme, Haarlem, The Netherlands), 0.1 M NaH_2_PO_4_, 0.01 M EDTA, 0.1% Triton-X, and 0.5 mM potassium ferri- and ferrocyanide at 37 °C until the appearance of a blue color. Stained seedlings were sequentially cleared in 70% and 100% ethanol for 2 h each. Samples were exposed to 10% (*v*/*v*) glycerol/50% (*v*/*v*) ethanol and 30% (*v*/*v*) glycerol/30% (*v*/*v*) ethanol in the final step of dehydration. GUS stained images were photographed under a microscope (Leica, Wetzlar, Germany). Seven-week-old stems were sectioned and incubated with phloroglucinol-HCl (phloroglucinol in ethanol was saturated in 20% HCl) solution for histochemical staining of lignin, as reported previously [4,5,11].

### 4.4. Total RNA Isolation and Real-Time Quantitative PCR (qPCR)

Total RNA extraction was performed using Pure link^TM^ RNA Mini Kit (Invitrogen, Carlsbad, CA, USA) according to the manufacturer’s instructions, and the RNA quality was vitalized by agarose gel loading (Appendix A). The extracted total RNA was quantified using a spectrophotometer (Nano-MD UV-Vis, Scinco, Seoul, Korea). RevertAid Reverse transcriptase (Thermo, Waltham, MA, USA) was used in a 20 μL reaction volume to synthesize the complementary DNA (cDNA). Real-time quantitative PCR (qPCR) was performed using TB Green™ Premix Ex Taq™ (Takara, Shiga, Japan) and Thermal Cycle Dice real-time PCR system (Takara, Shiga, Japan) as previously described [4]. To determine the relative fold-differences in template abundance for each sample, the Ct value for each of the analyzed genes was normalized to the Ct value for *β-actin* (At5g09810) and was calculated relative to a calibrator using the formula 2-ΔΔCt. Three independent experiments were performed for each primer set (Appendix A). The primer efficiency was determined according to the method of Livak and Schmittgen [25] in order to validate the ΔΔCt method. The gene-specific primer sequences for the target genes are listed in Appendix A.

### 4.5. Acetyl Bromide Soluble Lignin Assay for Total Lignin Quantification

Acetyl bromide assay to determine lignin content was performed according to a previously reported protocol [26]. The whole primary stems from 7-week-old plants were ground with liquid nitrogen and lyophilized for 48 h. Lyophilized powders were then filtered through a 425 μm screen. Each sample (10 mg) was washed four times with 95% (*v*/*v*) EtOH and twice with distilled water to remove soluble components. After air-drying at 60 °C for 12 h, products were dissolved in 2 mL of 25% (*v*/*v*, in glacial acetic acid) acetyl bromide and incubated at 70 °C for 30 min, to prevent excessive carbohydrate degradation that can distort the absorption spectrum. After incubation, 0.9 mL of 2 M NaOH, 3 mL of acetic acid, and 0.1 mL of 7.5 M hydroxylamine HCl were sequentially added, followed by centrifugation at 4000× *g* for 10 min. The supernatant was diluted 20-fold with glacial acetic acid, and absorbance was measured at 280 nm using a spectrophotometer (Nano-MD UV-Vis, Scinco, Seoul, Korea). Acetyl bromide soluble lignin (%) was calculated according to the previous report [27].

### 4.6. Quantification of Cellulose Content

Cellulose quantification in Arabidopsis was conducted according to a previously reported protocol [28]. Specifically, 50 mm sized stem segments were collected from 8-week-old plants and sequentially treated with 70% (*v*/*v*) EtOH and acetone followed by air-drying at 37 °C. The mass of the samples, which is the weight of cell wall materials, was measured before further processing. The samples were boiled in acetic/nitric reagent (acetic acid:nitric acid:water = 8:1:2) for 30 min to remove hemicellulose and lignin. They were then washed with water before being treated with acetone and then air-dried at 37 °C. Thereafter, samples were treated with 67% (*w*/*v*) H_2_SO_4_ for cellulose breakdown to produce monomeric sugars, which were measured by spectrophotometer (Nano-MD UV-Vis, Scinco, Seoul, Korea) at 620 nm using 0.3% anthrone (*w*/*w*, in concentrated H_2_SO_4_) as a dye. The amount of glucose in the samples was calculated based on the standard curve of D-glucose. Cellulose content (% cell wall) was calculated using the formula: Cellulose content (% cell wall) = amount of glucose in the sample (µg)/cell wall weight (µg) × total volume of H_2_SO_4_ used in the anthrone assay (µL).

### 4.7. Diaminobenzidine (DAB) Staining

DAB staining was used for in situ H_2_O_2_ detection. The cut stems were immersed in DAB staining solution containing 1 mg/mL DAB (bioWORLD, Dublin, OH, USA), 10 mM Na_2_HPO_4_, and 0.05% (*v*/*v*) TWEEN 20, infiltrated in a vacuum for 10 min and then incubated with shaking at 80 rpm for 4 h. The stained stems were boiled with a bleaching solution (ethanol:acetic acid:glycerol, 3:1:1) at 90 °C for 15 min to bleach out the chlorophyll after incubation. Images were taken using a microscope (DM3000 LED, Leica, Wetzlar, Germany).

## Figures and Tables

**Figure 1 plants-11-00200-f001:**
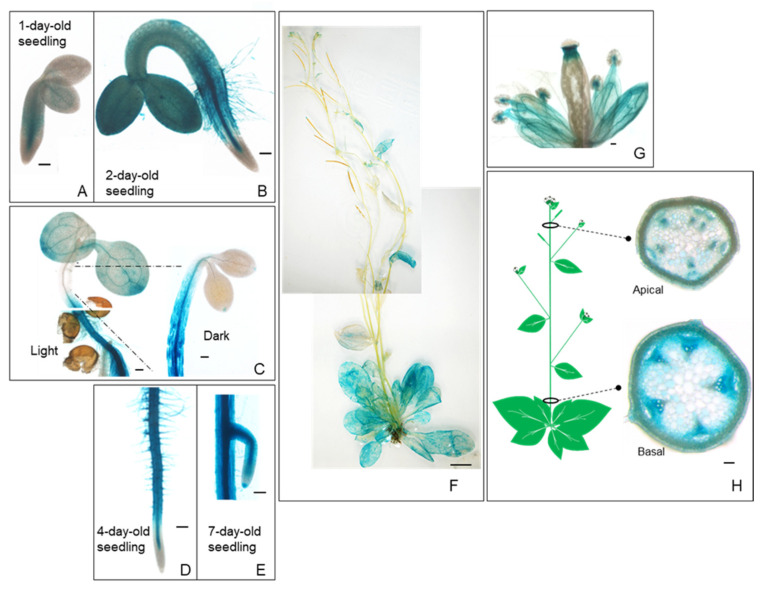
Spatial and temporal expression patterns of *pPLAIIIγ* gene in Arabidopsis. Histochemical analysis of GUS expression harboring *PropPLAIIIγ*::*GUS* at different developmental stages. (**A**) 1-d-old seedling. (**B**) 2-d-old seedling. (**C**) Aerial part of 4-d-old seedlings grown under long day (16 h light/8 h dark) and dark conditions. (**D**) Root of 4-d-old seedlings. (**E**) The meristematic zone of the lateral root in 7-d-old seedlings. (**F**) 6-week-old whole plant. (**G**) Floral organs. (**H**) The cross-sections show the GUS expression in the vasculature of the apical and basal parts of the stem. Scale bars = 100 μm (**A**–**E**,**G**,**H**) and 1 cm (**F**).

**Figure 2 plants-11-00200-f002:**
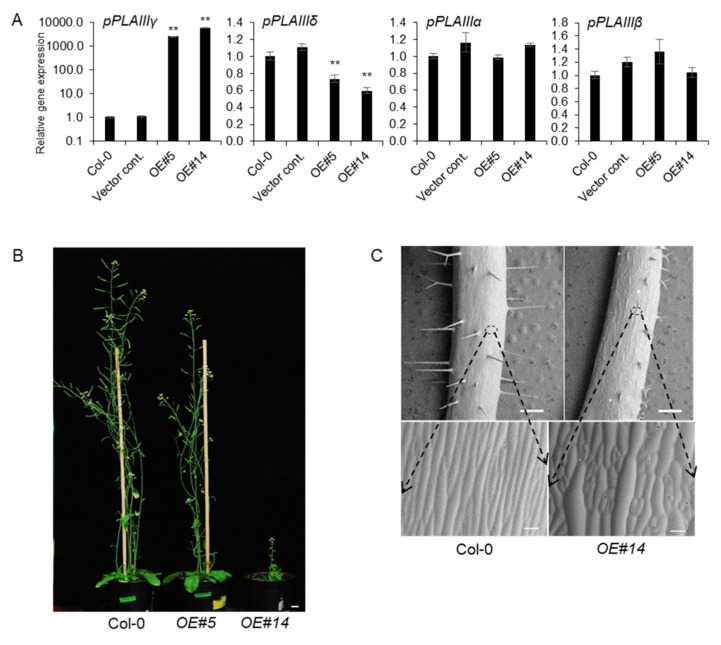
*pPLAIIIγOE* shows inhibited longitudinal growth and altered elongation pattern of epidermal cells. (**A**) Expression levels of each *pPLAIII* gene in 2-week-old seedlings of controls and *pPLAIIIγOE*. Each data point represents the average ± SE of three independent replicates at *p* < 0.01 (**), respectively. (**B**) Growth phenotype of 7-week-old Col-0 and *pPLAIIIγOE*. Scale bars = 1 cm. (**C**) Epidermal cell growth patterns are altered in the *pPLAIIIγ*OE lines. All images were taken using a low-vacuum scanning electron microscope (JSM-IT300, JEOL, Seoul, Korea) at 10.8 mm working distance and 20.0 kV.

**Figure 3 plants-11-00200-f003:**
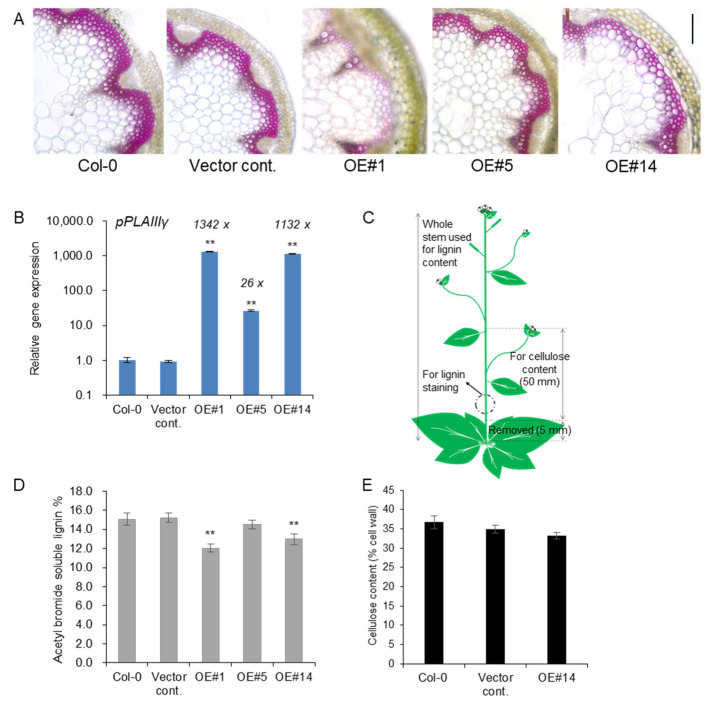
Lignin content was reduced in stem of *pPLAIIIγOEs* without altering cellulose content. (**A**) Phloroglucinol-HCl staining and (**B**) transcript levels of *pPLAIIIγ* in controls and OE lines (**C**) Arabidopsis image showing the parts used for lignin and cellulose analysis. (**D**) Total content of lignin by acetyl bromide method, and (**E**) quantification of cellulose content in 7-week-old stems of controls and *pPLAIIIγOE*. Each data point represents the average ± SE of multiple independent replicates at *p* < 0.01 (**), respectively. n = 3 (**B**), n = 4 (**D**), and n = 5 (**E**).

**Figure 4 plants-11-00200-f004:**
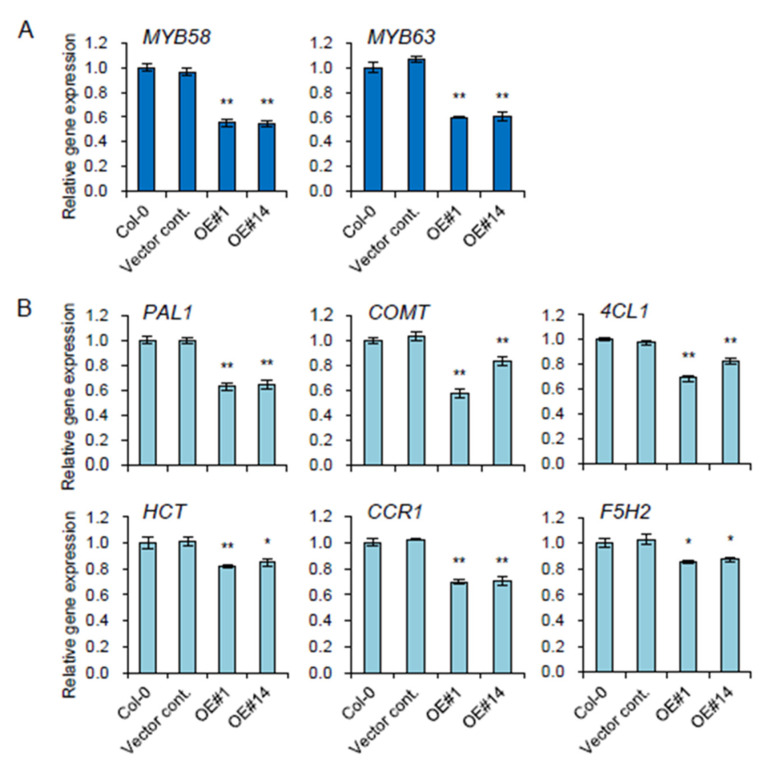
Genes involved in lignin biosynthesis were downregulated in the stem of *pPLAIIIγ*OE lines. (**A**) Two transcriptional activators, *MYB58* and *MYB63*, and (**B**) genes involved in lignin biosynthesis were quantified by qPCR. Each data point represents the average ± SE of three independent replicates at *p* < 0.05 (*) and *p* < 0.01 (**), respectively.

**Figure 5 plants-11-00200-f005:**
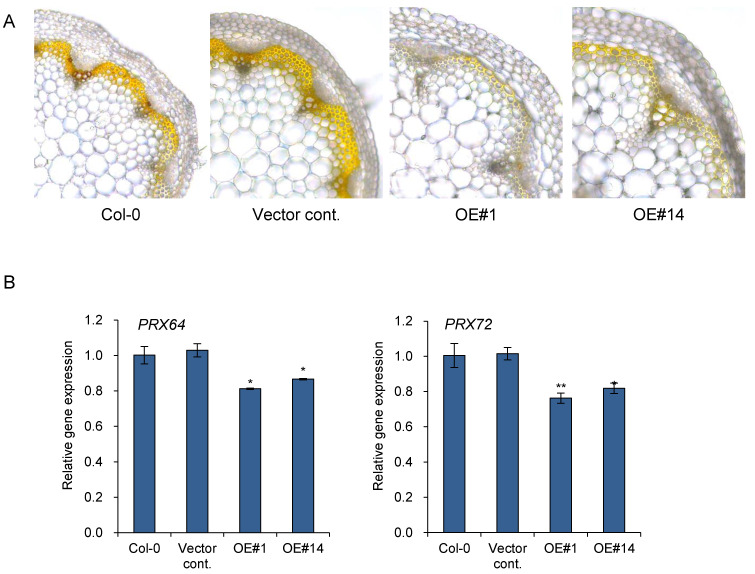
Accumulation of hydrogen peroxide was reduced in *pPLAIIIγOE* lines. (**A**) 3,3′-diaminobenzidine (DAB) staining and (**B**) expression levels of two peroxidases, *PRX64* and *PRX72*, in 7-week-old stem. Scale bar = 10 μm. Each data point represents the average ± SE of three independent replicates at *p* < 0.05 (*) and *p* < 0.01 (**), respectively.

## Data Availability

All the data included in this study have been presented within the manuscript (Figure 1, Figure 2, Figure 3, Figure 4 and Figure 5) and/or as Appendix A.

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
