# Peer review of "Overexpression of pPLAIIIγ in Arabidopsis Reduced Xylem Lignification of Stem by Regulating Peroxidases"

_plants, 2022, doi:10.3390/plants11020200_

Round 1

Reviewer 1 Report

The manuscript entitled “Overexpression of pPLAIIIγ in Arabidopsis Reduced Xylem Lignification of Stem by Regulating Peroxidases” by Jin Hoon Jang, Hae Seong Seo, Ok Ran Lee, is devoted to characterization of previously obtained pPLAIIIγ line in the frame of formation of dwarf plants with altered anisotropic growth and reduced lignification of stem. It is an interesting study. However, I have several questions to answer:

  • The authors provided no data about the quality of RNA used for gene expression study. The authors should provide (for instance, in supplementary materials) the picture of agarose gel-electrophoresis of RNAs used for gene expression experiments in order to prove the quality of RNA used for gene expression experiments.
  • The quality of some pictures is low. For instance, the quality of pic. 1 is low. It is hard to interpret what is exactly depicted on the figure 1F and 1H.
  • What was the reference gene or genes to study gene expression of pPLAIIIγ::GUS? How the authors selected particular these reference genes out of dozen ones? Why the authors consider that the selected genes are stable expressing in plant cells?
  • Did the authors perform an assessment of the effectiveness of the amplification for target and reference genes?

Author Response

The manuscript entitled “Overexpression of pPLAIIIγ in Arabidopsis Reduced Xylem Lignification of Stem by Regulating Peroxidases” by Jin Hoon Jang, Hae Seong Seo, Ok Ran Lee, is devoted to characterization of previously obtained pPLAIIIγ line in the frame of formation of dwarf plants with altered anisotropic growth and reduced lignification of stem. It is an interesting study. However, I have several questions to answer:

[1] The authors provided no data about the quality of RNA used for gene expression study. The authors should provide (for instance, in supplementary materials) the picture of agarose gel-electrophoresis of RNAs used for gene expression experiments in order to prove the quality of RNA used for gene expression experiments.

[Response] As you requested, we show the RNA quality checked by agarose gel loading as below. It is also incorporated as supplementary Figure 3 (Figure S3). Relevant sentence under “4.4 Total RNA isolation and real-time PCR” was revised as below,

Line 286-288: Total RNA extraction was performed using Pure linkTM RNA Mini Kit (Invitrogen, Carlsbad, CA, USA) according to the manufacturer’s instructions, and the RNA quality was vitalized by agarose gel loading (Figure S3).  

[Fig image]

[2] The quality of some pictures is low. For instance, the quality of pic. 1 is low. It is hard to interpret what is exactly depicted on the figure 1F and 1H.

[Response] Figure 1F displays whole plant to show overall GUS staining where you can see the expression of rosette leaves, cauline leaves, and in floral organs. The more detailed GUS expression in floral organ is in Figure 1G. Figure 1H images are especially cross-sectioned cut of basal and apical part of stem as depicted in left side of Arabidopsis image. GUS staining was observed in the vasculature (xylem and phloem) of the stem.

For your better understanding, Figure 1H legend is revised as below,

Line 363-364: (H) The cross-sections show the GUS expression in the vasculature of the apical and basal parts of the stem.

If you magnify each image it would be much better since the quality of original images submitted is fine. In the process of showing all the large images, the picture image seems to get smaller and looked low quality.

However, to provide still the better image, we replaced Fig 1F and 1H images.

[3] What was the reference gene or genes to study gene expression of pPLAIIIγ::GUS? How the authors selected particular these reference genes out of dozen ones? Why the authors consider that the selected genes are stable expressing in plant cells?

[Response] The GUS reporter system (GUS: β-glucuronidase) is a widely used reporter gene system, particularly useful and well-established in plant. As mentioned in the main text, “to gain insights into the temporal and spatial distribution patterns of pPLAIIIγ transcripts ~”, GUS reporter gene was used. It is a commonly used method for histochemical analysis. As described in Method section, it is a fusion construct with promoter sequence of pPLAIIIγ fused with GUS gene. To clearly specify, relevant naming was revised into PropPLAIIIα::GUS in whole MS.

Promoter fusion construct with GUS was stably transformed into Arabidopsis plant and the GUS histochemical analysis was performed using transformants as described in Method section.

For your better understanding, check it up the below journal

Jefferson, RA et al., GUS fusions: β-glucuronidase as a sensitive and versatile gene fusion marker in higher plants. The EMBO Journal6 (13): 3901–7.

[4] Did the authors perform an assessment of the effectiveness of the amplification for target and reference genes?

[Response]Of course, it was definitely done as it should have been accompanied by. To determine the relative fold-differences in template abundance for each sample, the threshold cycle (Ct) value of each gene-specific genes was normalized to that of β-actin, and the formular 2△△Ct was used to calculate the gene expression levels. As we already cite our previous work [4] in the MS, the details are well described in [4], and we further elaborated in method section for clear understanding by addressing the theory as below,

Line 295-300: Thermal Cycle Dice real-time PCR system (Takara, Shiga, Japan) as previously described [4]. To determine the relative fold-differences in template abundance for each sample, the Ct value for each of the analyzed genes was normalized to the Ct value for β-actin (At5g09810), and was calculated relative to a calibrator using the formula 2 -ΔΔCt. Three independent experiments were performed for each primer sets (Table 1). The primer efficiency was determined according to the method of Livak and Schmittgen [25] in order to validate the ΔΔCt method.

More information as below for your better understanding.

1)The deltaCT is calculated; (CT value of the target gene – CT value of b-actin gene), and 3 times of repeat were conducted.

2)For 2△△Ct method, we took the theory from Livak and Schmittgen [19] as below,

“For the ΔΔCT calculation to be valid, the amplification efficiencies of the target and reference must be approximately equal. If the absolute value of the slope is close to zero, the efficiencies of the target and reference genes are similar, and the ΔΔCT calculation for the relative quantification of target may be used.”

3)Rea-time qPCR experiments was performed after validation of the amplification curve and dissociation curve in the Takara Dice Real time single program as you see in the below.

Reviewer 2 Report

The manuscript by Jang et al. tries to put some information and results by involving the overexpression of gene pPLAIIIγ in plant Arabidopsis. This overexpression led to the formation of dwarf plants with altered anisotropic growth and reduced lignification of the stem. This was also supported by checking the transcript levels of lignin biosynthesis-related genes as well as lignin-specific transcription factors. This also put forth a link about the functional role of peroxidase in the reduction of lignification by this pPLAIIIγ gene.

My comments and questions to authors are as follows:

The introduction of the paper is less and can be elaborated a bit more, describing in a bit more detail the H2O2 part?

How do the authors explain the enormous differences in the growth of two overexpression lines (Line 5 and Line 14) in figure 2B? Why do they differ so much from each other?

Why did not authors provide the scanning electron microscope images of line 5?

What about the cellulose content in the cell wall of other lines 1 and 5? Figure 3E

What about the gene expression of lignin biosynthesis genes in line no. 5?

In figure no. 5A  when in situ detection of H2O2 is shown by staining with 3,3- diaminobenzidine (DAB), the number of cells stained in the region of the sections is also less in the OE lines. In simple terms, I see fewer cells in the region when compared to controls. This factor might also implicit the less uptake of the stain may be correlated proportionally with these fewer cells. Some thing to take into consideration in your manuscript.

Lastly, It is not always recommended in journals but in my observation, it is the best strategy to use numbers to each line in the manuscript to highlight or point to the comments or questions during reviewing.

Some minor comments:

Change text size at the end of section 4.3. in material and methods,

g is in italic in section 4.5. when using centrifugation

GUS- full form

Author Response

The manuscript by Jang et al. tries to put some information and results by involving the overexpression of gene pPLAIIIγ in plant Arabidopsis. This overexpression led to the formation of dwarf plants with altered anisotropic growth and reduced lignification of the stem. This was also supported by checking the transcript levels of lignin biosynthesis-related genes as well as lignin-specific transcription factors. This also put forth a link about the functional role of peroxidase in the reduction of lignification by this pPLAIIIγ gene.

My comments and questions to authors are as follows:

[1] The introduction of the paper is less and can be elaborated a bit more, describing in a bit more detail the H2O2 part?

[Response] To help for a smooth flow of understanding, the below red characters were incorporated in “Introduction” part. More detailed explanation seems to be well-addressed in ‘Result and Discussion part’.

Line 83-85: Oxidation of monolignols is required in the final stage of lignin polymerization, and peroxidases (PRXs) are involved in the random cross-linking to facilitate lignin polymerization [13]. Two PRXs, PRX64 and PRX72, function

[2] How do the authors explain the enormous differences in the growth of two overexpression lines (Line 5 and Line 14) in figure 2B? Why do they differ so much from each other?

[Response] There are huge differences in mRNA levels of each transformants. Line 5 is a moderate line compared to that of controls but the Line 14 (Figure 2B and Figure 3B), and Line 1 (Figure 3B) are both highly expressing strong lines as indicated by the transcript-fold levels. The level of mRNA is indicated in Fig 3B (stem) where Line 5 is just 26-fold whereas, it is 1,342- and 1,132-fold in Line 1 and 14, individually. In Figure 2B (seedling), if you look at the Y-axis interval, there are also huge difference. It was 2,580-fold in Line 5 and 5,768-fold in Line 14 when I double-checked our raw excel chart, and it is already addressed in the relevant text. This kind of huge difference was also observed in our previous reports [3-5, 11,15] using constitutive 35S CaMV promoter. The more transcripts, the more severe phenotype we observed as expected, which is also generally reported phenotypes in this kind of overexpression study. Moderately or weakly expressing line do not show significant different phenotype. That’s why we analyzed several one copy inserted lines following Mendelian segregation ratio (1:3) to characterize transgenic lines.

 [3] Why did not authors provide the scanning electron microscope images of line 5?

[Response] As I answered above concerning the transcript levels and corresponding phenotype, Line 5 where the gene is expressing weak, showed weaker phenotype. Thus, we analyzed Line 14 and more other Line 1 to confirm the observed phenotype is general for the strongly OE lines. The image was taken by a light microscope (Leica, as described in method section) after DAB staining stem cuts.

[4] What about the cellulose content in the cell wall of other lines 1 and 5? Figure 3E.

[Response] In the case of OE#5, the phenotype and lignin content were also not significantly different compared to that of WT. Our previous study (Jang and Lee 2020 Plants 9(4):451, Jang and Lee 2020 J. Ginseng Res. 44(2) 321-331.), where Arabidopsis pPLAIIIα and ginseng pPLAIIIβ were overexpressed, also reported similar results where only the lignin content decreased without altering the cellulose content. Based on these supporting results, only Line 14, which showed strong overexpression was used to analyze the cellulose content only to obtain the similar pattern of outcomes with other pPLAIII-OE lines. We do more focused on the level of lignin by overexpression pPLAIIIγ.

[5] What about the gene expression of lignin biosynthesis genes in line no. 5?

[Response] We do more focused on Line 14 and 1 which showed general pPLAIII-OE phenotypes. Since the lignin content was not significantly different in Line 5 (Figure 3D, further qPCR was NOT necessarily required.

[6] In figure no. 5A when in situ detection of H2O2 is shown by staining with 3,3- diaminobenzidine (DAB), the number of cells stained in the region of the sections is also less in the OE lines. In simple terms, I see fewer cells in the region when compared to controls. This factor might also implicit the less uptake of the stain may be correlated proportionally with these fewer cells. Something to take into consideration in your manuscript.

[Response] As your comment, it is true that the number of cells present in the DAB stained area is less in the OE line than in the WT as shown in the figure below. (Cell number only in blue box- Col-0: 46, Vector cont.: 53, OE#1: 23, OE#14: 22). You were concerned that a small number of cells could lead to less uptake of the DAB stain. However, looking at the DAB staining results of Piovesana et al (DOI: https://doi.org/10.1101/2021.08.16.456532), in the case of crk10-A397T mutant, the cell size is larger than that of WT, so the number of cells per unit area is small, but it can be seen that the DAB stained area is darker than that of WT. Based on this, it can be seen that cell density is not significantly related to less uptake of the stain. Therefore, it is not considered that the result of figure. 5A is distorted.

(In Figure 5A of this paper)

(In Figure 3 of Piovesana et al.)

Additionally, we have previously reported that that overexpression of other close homolog, pPLAIIIa, also showed enlarged cells but the number of cells per the whole cross-sectioned area of stem was similar with WT. No matter the cell size, cell number is not significantly different in controls and OE lines. Concurrent lignin staining and absolute lignin quantification was showed reduction.

Lastly, It is not always recommended in journals but in my observation, it is the best strategy to use numbers to each line in the manuscript to highlight or point to the comments or questions during reviewing.

 [Response]As your comment, it is revised by addressing Line No and pinpointing the comments.

Some minor comments:

Change text size at the end of section 4.3. in material and methods,

 [Response]As your comment, it is double-checked. It seems likely (sometimes) that the font is unexpectedly changed during converting/submitting MS.

g is in italic in section 4.5. when using centrifugation

[Response]You are perfectly right. As your comment, it is revised.

GUS- full form

[Response]As your comment, GUS (β-Glucuronidase) is revised for the title of 4.3 and corresponding main text under Result section 2.1.

Round 2

Reviewer 1 Report

I would like to thank the authors for the answers.I believed that the introduced information highly improved the manuscript for understanding it more easily.

Author Response

I would like to thank the authors for the answers.I believed that the introduced information highly improved the manuscript for understanding it more easily.

[Response]Thank you!

Reviewer 2 Report

I appreciate the reply of the authors to my query and comments. The manuscript has been upgraded as was suggested. I would just like to mention to the authors that the submitted new version has missing figures as was in the earlier version initially of course this might have been done mistakenly. Can the authors resubmit the manuscript with figures where changes were made so that we can correlate the new changes made in this version before I accept the manuscript for publication in this journal.  

Author Response

I appreciate the reply of the authors to my query and comments. The manuscript has been upgraded as was suggested. I would just like to mention to the authors that the submitted new version has missing figures as was in the earlier version initially of course this might have been done mistakenly. Can the authors resubmit the manuscript with figures where changes were made so that we can correlate the new changes made in this version before I accept the manuscript for publication in this journal.  

[Response] I have attached converted ver of MS. Thank you!
